# Development of a Mating Disruption Program for a Mealybug, *Planococcus ficus*, in Vineyards

**DOI:** 10.3390/insects11090635

**Published:** 2020-09-16

**Authors:** Kent M. Daane, Glenn Y. Yokota, Vaughn M. Walton, Brian N. Hogg, Monica L. Cooper, Walter J. Bentley, Jocelyn G. Millar

**Affiliations:** 1Department of Environmental Science, Policy, and Management, University of California, Berkeley, CA 94720-3114, USA; gyyokota@gmail.com; 2Department of Horticulture, Oregon State University, Corvallis, OR 97331, USA; vaughn.walton@oregonstate.edu; 3USDA-ARS, Invasive Species and Pollinator Health Research Unit, Albany, CA 94710, USA; brian.hogg@usda.gov; 4University of California Cooperative Extension, 1710 Soscol Avenue, Napa, CA 94559, USA; mlycooper@ucanr.edu; 5Kearney Agricultural Center, University of California IPM Program, Parlier, CA 93648, USA; wjbentley@ucanr.edu; 6Department of Entomology, University of California, Riverside, CA 92521, USA; jocelyn.millar@ucr.edu

**Keywords:** biological control, Hemiptera, seasonal development, semiochemicals, sex pheromones, sustainable agriculture, vineyard pests

## Abstract

**Simple Summary:**

The vine mealybug is a key insect pest of vineyards that currently is controlled by one or more insecticide applications per season. Here, we sought to develop a more sustainable control tool by using the mealybug’s sex pheromone to reduce mating and thereby lower pest damage. The mature female mealybug emits a sex pheromone that the winged adult male uses to find and mate with females. Synthetically produced sex pheromone, specific to the vine mealybug, was enclosed in commercial dispensers and deployed in vineyards in 2004–2007 studies to determine if mating disruption could provide a viable control option. Trials were conducted in commercial vineyards with cooperating farmers. Across all trials, mating disruption reduced pheromone trap captures of adult male mealybugs—an indication that the population numbers were lowered—and there was often a reduction mealybug numbers on vines and/or crop damage. There was not a clear reduction in the proportion of female mealybugs with ovisacs (a cottony-like mass containing mealybug eggs), but this may have resulted from the production of non-viable ovisacs that were not differentiated in the field samples. Pheromone trap captures were never lowered to zero (often called trap shut down), possibly because trials were conducted in vineyards with unusually high mealybug densities. Trap capture patterns commonly began low in April-May, increased in mid-July or August, and often decreased in September–October when post-harvest insecticides were applied. Results over all years suggest season-long coverage or late season coverage may be as or more important than dose per hectare. This research was used to help initiate the commercialization of mating disruption products for the vine mealybug, which are now being successfully used throughout the world’s grape-growing regions where this pest is found.

**Abstract:**

The vine mealybug (VMB), *Planococcus ficus* (Hemiptera: Pseudococcidae), is a key insect pest of vineyards, and improvements in sustainable control of this pest are needed to meet increasing consumer demand for organically farmed products. One promising option is mating disruption. In a series of experiments conducted from 2004 to 2007, we tested the effects of mating disruption on trap captures of *Pl. ficus* males in pheromone-baited traps, on *Pl. ficus* numbers and age structure on vines, and on damage to grape clusters. From 2004 to 2005, the effects of dispenser load (mg active ingredient per dispenser) were also assessed, and dispensers were compared to a flowable formulation. Across all trials, mating disruption consistently reduced pheromone trap captures and often reduced mealybug numbers on vines and/or crop damage, regardless of the pheromone dose that was applied. Reductions in *Pl. ficus* densities in mating disruption plots were not accompanied by clear effects on mealybug population age structure; however, production of non-viable ovisacs by unmated females may have obscured differences in proportional representation of ovisacs. Pheromone trap captures were never lowered to zero (often called trap shut down), possibly because trials were conducted in vineyards with unusually high *Pl. ficus* densities. Trap-capture patterns in both treated and control plots commonly began low in April–May, increased in mid-July or August, and often decreased in September–October when post-harvest insecticides were applied. During the four-year trial, the release rate from plastic sachet dispensers was improved by industry cooperators as pheromone was released too quickly (2004) or not completely released during the season (2005–2006). The flowable formulation performed slightly better than dispensers at the same application dose. Results over all years suggest season-long coverage or late-season coverage may be as or more important than dose per hectare. Development of a dispenser with optimized season-long pheromone emission or targeted seasonal periods should be a future goal.

## 1. Introduction

Over the past 100 years, a series of mealybug (Hemiptera: Pseudococcidae) species have attacked California vineyards, with five species currently causing damage and a sixth species posing a threat [1]. These include the native grape mealybug, *Pseudococcus maritimus* (Ehrhorn), and the invasive long-tailed mealybug, *Ps. longispinus* (Targioni Tozzetti), obscure mealybug *Ps. viburni* (Signoret), *Planococcus ficus* (Signoret), *Ferrisia gilli* Gullan, and most recently, the pink hibiscus mealybug, *Maconellicoccus hirsutus* (Green). Of these, *Pl. ficus* is the dominant and most problematic mealybug for California vineyard managers (Figure 1). Similar to most other vineyard mealybugs, *Pl. ficus* can feed on the vine’s trunk, canes, leaves, or berry clusters; however, its fast development rate, cryptic feeding locations, temperature tolerances, and high population densities elevate its pest status in California [2]. Feeding damage can result in defoliation; repeated annual infestations severely weaken vines, and untreated populations can result in vine death [3]. For table grapes, any live or dead mealybugs and the accumulated honeydew and associated sooty molds will cause cosmetic damage to the grape cluster and reduce its marketability [2]. The transmission of viruses is an added concern, especially among wine grape growers [4,5,6]. Grapevine leafroll disease (GLD) is caused by a complex of several viruses [7,8,9,10], collectively known as grapevine leafroll-associated viruses (GLRaVs) (Figure 1). Especially in cool-climate regions, the pathogen can be damaging to vine health, crop yield, and wine quality [11]. GLRaV-3 is the predominant species in most California vineyards [7], and its vector-driven disease spread by different mealybug species has been confirmed worldwide [12,13,14,15,16,17,18].

*Planococcus ficus* is one of the more widespread mealybugs damaging vineyards, being found in California, Mexico, South America, Europe, the Middle East, and South Africa [19]. For many vineyard mealybug species, natural enemies are able to suppress pest densities to acceptable levels [20]; for example, *Ps. viburni* was brought under control in New Zealand by the parasitoid *Pseudaphycus maculipennis* Signoret (Hym.: Encyrtidae) [21]. A number of natural enemies attack *Pl. ficus*, most importantly the species complex of *Anagyrus* nr sp. *pseudococci* Signoret and lady beetles such as the mealybug destroyer, *Cryptolaemus montrouzieri* Mulsant [22,23,24,25]. Nevertheless, pesticides are commonly needed to keep *Pl. ficus* below damaging levels. Historically, the pesticides applied included chlorinated hydrocarbons (e.g., DDT) and organophosphates (e.g., parathion), but these have largely been replaced by newer materials with different modes of action, such as neonicotinoids, insect growth regulators, botanicals, and biosynthesis inhibitors [26,27,28,29]. For organic or sustainable farming programs, neem, light mineral oils, lime-sulfur, citrus products, and fatty acid soaps have been used. However, studies of these products have provided mixed efficacy results [30], and some of these materials can disrupt natural enemy activity [31,32,33].

Improvements to *Pl. ficus* control programs are needed to meet the increasing public demand for more sustainable farming practices [34,35]. The uses of semiochemicals for monitoring and mating disruption (MD) are considered key tactics for vineyard sustainability and fit well in areawide pest management approaches, such as the areawide control in Italy of the European grapevine moth, *Lobesia botrana* (Denis & Schiffermüller) (Lepidoptera: Tortricidae) [36,37]. It was known that mature female *Pl. citri* emit a sex pheromone to attract the winged adult males [38], and this pheromone, initially identified as (+)-(1*R*,3*R*)-2,2-dimethyl-3-(1-methylethenyl)cyclobutanemethanol acetate, can be synthesized [39]. With this information as a starting point, a semiochemical approach for *Pl. ficus* was initially focused on developing a monitoring tool for this invasive pest in California, which can be difficult to visually monitor because of its clumped distribution [40] and its location in protected parts of the vine, such as the trunk [41]. To improve monitoring programs, sex pheromones for California mealybug species were investigated for *Pl. ficus* [42], *Ps. viburni* [43], *Ps. maritimus* [44], and *Ps. longispinus* [45], and a series of studies showed that trap counts can be correlated to population density and crop damage [18,46,47,48].

*Planococcus ficus* sex pheromone as a lure for pheromone traps was available in 2006 from Suterra LLC (Bend, OR, USA). Even before the pheromone was commercially available as a lure, researchers investigated its use for MD, whereby vineyards are permeated with synthetic sex pheromone to reduce or delay mating. Positive results from initial MD trials using a flowable, microencapsulated formulation [49] led to issuance of a ‘Section 18′ emergency exemption (United States Environmental Protection Agency) for large scale experimentation throughout California using CheckMate^®^ plastic dispensers (Suterra LLC, Bend, OR, USA) loaded with 150 mg active ingredient. Since then, a number of commercial products for *Pl. ficus* MD have been tested worldwide, including in Tunisia [50], Israel [51,52], and Italy [53,54]. Here, we describe the initial *Pl. ficus* mating disruption research trials (2004–2007) in California vineyards that eventually led to a commercial product. Areas of investigation included effects of dispenser load, release rate, and pheromone purity; comparison of discrete dispensers to flowable formulations; male mealybug flight patterns at the vineyard and landscape scale. Our goals were to determine how mating disruption fits within an integrated program and to develop recommendations for commercial use of the product as a management tool for *Pl. ficus*.

## 2. Materials and Methods

### 2.1. Replicated Field Trials—2004 and 2005

In 2004 and 2005, replicated field trials focused on assessing the effects of dispenser load and comparing discrete dispensers to a flowable formulation. To reduce among-plot variation, trials were conducted using a randomized block design, and in similarly managed vineyards located near Del Rey, California (Fresno County). The vineyards were ~15 year-old Thompson Seedless *cv.*, with vines trained to a single- or double-T trellis and a clean vineyard floor maintained by running a disc over the row middles and applying the herbicide Roundup^®^ (Monsanto Co., Creve Coeur, MO, USA) to the berms. All vineyards were within a ~500 ha area. During the trials, plastic sachet and plastic tube (rope) dispensers were used, which were hand-clipped or twist-tied to the vine, respectively; these dispensers have a reservoir of pheromone that allow for longer residual activity (120 to 180 days). A microencapsulated flowable formulation was also used where the pheromone was enclosed in a polymer capsule that is small enough to be mixed with water and applied via an airblast sprayer. This flowable formulation can be ‘tank-mixed’ with other materials, such as pesticide(s), which is attractive for farm operations. One disadvantage is that this flowable formulation had a residual impact of 4–6 weeks [49]; therefore, for seasonal-long activity, multiple applications are needed. It was not the intent of these original studies to compare products, but to provide proof of concept with the goal of government approval for commercialization.

In the first trial (2004 only), two loads of an Isomate rope dispenser (ShinEtsu, Vancouver, WA, USA) were tested; Isomate dispensers were loaded at 60 mg per dispenser, and the two rates were created by using the full 20 cm dispenser or cutting dispensers into three equal parts, resulting in 20 mg per dispenser. The trial was conducted in four vineyards, with each receiving the two MD rates and a control, established in a randomized block design (RBD) and with treatment plots ranging from ~1–4 acres, dependent on the collaborating growers’ land donation (all treatment plots at each vineyard were the same size). There was a 20-row buffer between plots in each vineyard. Dispensers were hung on every vine, however row and vine spacing varied among vineyards (1.8 × 3.7, 2.1 × 3.7, 2.4 × 3.7, and 2.1 × 3.3 m), and for this reason, dispensers were deployed at 935, 985, 990, and 1040 per ha, to create treatment rates of about 20 and 60 g (AI)/ha. Each site had considerable *Pl. ficus* populations and 3 of 4 vineyards received an application of Applaud^®^ (Nichino America, Wilmington, DE, USA) on either 19 or 26 May or 5 July, and in 2 of 4 vineyards a post-harvest application of Lorsban^®^ (Dow AgroSciences LLC, Indianapolis, IN, USA) was applied on either 9 or 11 October.

Adult male *Pl. ficus* flights were monitored, after Walton et al. [49], using three Pherocon Delta IIID sticky traps (Trécé Inc., Adair, OK, USA) per plot, each baited with a rubber septum lure loaded with 100 µg of the racemic pheromone (lavandulyl senecioate; Suterra LLC). Traps were hung from trellis wires such that they were positioned above the cordon but within the vine canopy and were placed in the two or three center rows of each plot, 15 to 50 vines from the edge. Traps were collected every two weeks from April to November and lures were replaced every 4 weeks. There was a total of 12 traps per monitoring period. Trapped insects were counted in the laboratory using a dissecting microscope. Crop damage was assessed near harvest time (5–6 August) using a 0–3 rating of sampled fruit, after Geiger et al. [55], where: 0 = no mealybug damage, 1 = a few mealybugs found or some honeydew present, 2 = fruit damage caused by mealybug and honeydew accumulation, but the cluster was salvageable by pruning or clipping off damaged sections, and 3 = severe damage and an unmarketable cluster. Economic damage was measured by rating clusters on 100 vines per plot, with vines selected at random. However, clusters above the trunk and, when possible, touching the trunk or cordon were preferentially selected as these clusters typically have a higher infestation level.

For the second trial (2004 and 2005), the following five treatments were established: Isomate dispensers (Scentry Inc., Billings, MO, USA) loaded at 7.5 mg and 15 mg per dispenser and applied at a rate of 555 per ha (4.15 and 8.3 g (AI)/ha, respectively); CheckMate Flowable^®^ (Suterra LLC), a microencapsulated formulation that was mixed with water (1 mL; 7.6 L) and applied using an airblast spray rig four times during the season at 8 g (AI)/ha per application (32 g/ha); CheckMate plastic dispensers (Suterra LLC) loaded at 60 mg per dispenser and applied at 555 per ha (33.3 g a.i./ha). The trials were conducted in five vineyards with measurable *Pl. ficus* damage to grape clusters the previous season. Vine × row spacing was similar in four vineyards (2.4 × 3.7 m) and denser in the fifth vineyard (1.8 × 3.7 m). In each vineyard, treatments were established (RBD) in plots that were five rows wide and separated from each other by 20–30 rows. Spacing of pheromone traps was similar among plots, but plot size varied from 0.16–0.3 ha (0.4–0.7 acres) as some blocks had longer rows and the spray application practices were the same through each row. All dispensers were deployed between 19–27 April 2004, with the flowable formulation applied on 20 April, 19 May, 16 June, and 19 July (about every 4 weeks). In 2005, this second trial was repeated, using the same plots and sampling design, with the exception that the treatment using Isomate dispensers loaded at 15 mg/dispenser was dropped and only three applications of the Flowable CheckMate formulation were applied (24 g (AI)/ha). At all vineyards, an application of Applaud was made in late May to early June of each year, and a post-harvest application of Lorsban was made in some blocks as determined by our grower collaborators.

In both 2004 and 2005, adult male *Pl. ficus* flight was monitored using three pheromone-baited traps per plot, and damage was assessed by rating clusters on 100 vines per plot (2500 vines total), as described previously. Additionally, new sampling methods were added to better assess treatment effect on *Pl. ficus* population dynamics. First, *Pl. ficus* densities were estimated on 10 randomly selecting vines in each plot by recording mealybugs found during a 5 min (2004) or 3 min (2005) search, conducted about every 2 weeks from May to September. A major objective of the timed counts was to determine if MD treatments affected *Pl. ficus* developmental stages or locations on the vine. During the timed search, an experienced sampler inspected the vine—under the bark of the trunk and cordon, basal leaves, and fruit—recording number of mealybugs by developmental stage (crawler or settled 1st instar, 2nd instar, 3rd instar, or immature female, adult female, and adult female with an ovisac) and location on the vine (ground to 30 cm above the soil line, trunk, cordon, cane, leaves, and fruit). Since this sampling was destructive a vine was only sampled once per growing season. Parasitized mealybugs were also recorded, determined by a mummified mealybug or an adult parasitoid exit hole [22]. Second, to determine treatment effect on egg production by adult females, from April to August we collected 20–40 adult females from each plot, isolating each in a 25 mm gelatin capsule for 30 or more days after which the number of emerged crawlers and parasitoids were recorded for each mealybug. Third, leaf samples were collected to determine the respective change in *Pl. ficus* density during the season relative to initial population density within each plot. To develop this assay, in April, vines were classified as having low infestation (no evidence of mealybug infestation), medium infestation (some evidence of mealybugs, such as blackened trunk from sooty molds, but <10 mealybugs found during a 5 (2004) or 3 (2005) minute search), and high infestation (ant activity, honeydew accumulation, blackened main trunks, and >10 live mealybugs during a 5 or 3 min search). Thereafter, these same vines were repeatedly sampled by randomly selecting 10 leaves near the vine crown and recording the number of mealybugs without removing the leaf (i.e., a nondestructive sample). There were five vines per plot (25 vines per treatment, 125 total) with mealybugs recorded every 2–3 weeks from May to August.

### 2.2. 2006 and 2007 Seasons

In 2006 and 2007, the effectiveness of CheckMate plastic dispensers was tested in six raisin vineyards located near Del Rey, California (Fresno County). The goal was to test dispensers under a more commercial system, using larger plot sizes and a single insecticide application in both the MD and control treatment. All vineyards were within the same ~500 ha area as the 2004 and 2005 studies previously described, and the blocks were similarly managed for raisin grapes. Vineyards were selected based on measurable *Pl. ficus* damage to grape clusters in the previous season. In each of the six vineyards, two 4 ha plots were assigned, separated by a buffer section of 30–60 rows. The two treatments, either MD and an insecticide or an insecticide only, were randomly assigned to each paired plot (six replicates). Pheromone dispensers loaded with 100 mg (AI) per dispenser were deployed at a rate of 617 per ha from 10–12 May 2006 and 7–10 May 2007 (61.7 g/ha). Each paired plot received the same insecticide application in each year. In 2006, three vineyards were treated with imidacloprid (Admire^®^, Bayer CropScience, St. Louis, MO, USA) between 3–6 June at 17.3 oz/ha, and three vineyards were treated with Applaud^®^ between 20–22 June at 29.6 oz/ha. In 2007, all six vineyards were treated with buprofezin (29.6 oz/ha) between 15–24 June.

Adult male *Pl. ficus* were monitored using two Pherocon Delta IIID sticky traps per plot (24 traps per monitoring period), and fruit damage was measured by rating clusters on 100 vines per plot at harvest time using the same 0–3 scale described previously. A variation of the timed count was used in 2006 and 2007 to reduce field sampling time. *Planococcus ficus* density was assessed on 100 randomly selected vines per plot using a 0–3 rating scale at four seasonal periods: May (pre-treatment), June, July, and August (harvest time). For each sample period, vines were searched for a 30 s period to assess *Pl. ficus* density using a presence/absence rating (2006) or a 0–3 scale, where: 0 = no mealybug damage, 1 = a few mealybugs found or some honeydew present, 2 = 5–10 mealybugs and honeydew accumulation, and 3 = > 10 mealybugs, considerable honeydew accumulation and unmarketable fruit. To determine parasitism rates, we collected 100 mealybugs (2nd instar to adult stages) in August, the primary period of parasitoid activity in the San Joaquin Valley [22], recorded their development stage and location by the level of refuge [41] as either located in an exposed (e.g., on leaves) or protected (e.g., under bark) location, and then isolated each in a gelatin capsule. After 2 months, the capsules were checked for dead or mummified mealybugs and emerged adult parasitoids, which were recorded by species and gender. For unparasitized mealybugs, the number of ovisacs produced was determined.

Additionally, in 2007, we placed traps in all (*n* = 58) commercial vineyards, regardless of whether they were in the test plots or not, across a 2 × 2.2 km grid (420 ha) that encompassed 4 of the 6 treatment vineyards (two of the tested vineyards were further to the south and not surrounded by other vineyards). Pheromone traps were deployed, as described previously, during the expected period of peak *Pl. ficus* flights (late July to early October); there were 1–2 traps per block, depending on size. The commercial vineyards were managed for raisin, juice, and table grapes and their management practices and insecticide applications varied and were not documented. Our goal was to determine if MD had an effect outside of the treated block, and to help determine if MD attracted male *Pl. ficus* from other vineyards, which would be indicated by a halo of higher counts surrounding the treated section.

### 2.3. Dispenser Load and Release

Concurrent with the 2004–2007 studies, each year we placed ~200 plastic CheckMate dispensers in vineyards at the Kearney Agricultural Research and Extension Center (Parlier, California, USA), typically in May, and collected 20 (2004) or 10 (2005–2007) dispensers either once per month (2004) or every 2–3 weeks (2005–2007). The collected dispensers were held in a freezer (−20 °C) until they could be delivered overnight to Suterra (Bend, OR location) to be analyzed for the amount of remaining synthetic pheromone, to determine if the sex pheromone was evenly released from the plastic dispensers during the season (May to November).

### 2.4. Pheromone Purity

Concurrent with the 2005 studies, the effect of the synthetic pheromone purity and load was compared. The pheromone (racemic lavandulyl senecioate) was prepared by Kuraray (Tokyo, Japan) at two purity levels (99 and 95% pure) and loaded into CheckMate dispensers (from two different production batches) at 100 mg (AI) per dispenser. The trial was conducted in Thompson Seedless vineyard blocks, managed for raisin grapes, as described previously, near Del Rey, CA. Plots were small, 5 rows × 30 vines (0.13 ha), set in a randomized complete block design (RBD) with four replicates. Treatment effect was assessed during the season by rating 50 vines as infested or clean in each plot (200 vines per treatment per sample date), and by a harvest-time assessment of fruit damage by rating clusters on 200 vines per plot using the 0–3 scale described previously.

### 2.5. Statistics

Results are presented as sample means ± SEM. Analyses were performed using Systat Software Inc. (version 13, San Jose, CA, USA). For the pheromone trap counts and visual mealybug counts, we compared season-long treatment effects using a Generalized Linear Model (GLM), with mealybug densities (trap counts, visual counts, percent infestation) as the dependent variable and treatment and date as the independent variables, with crossed treatment × date interaction. If more than two treatments were tested, Tukey Pairwise comparison was used to separate treatments. Data were transformed (log[x + 1] or square root [x + 1]) as needed to stabilize the variance. For categorical ratings of mealybug density and cluster damage, treatment effects were compared in a 2 by 2 contingency table with treatments separated using Pearson’s Chi-square test, with an experiment-wide error rate at *p* < 0.005 (α = 0.05/n, where n is the number of possible pairwise comparisons).

## 3. Results

### 3.1. Field Trials—2004 and 2005

The 2004 Isomate dispenser trial comparing two loads (20 and 60 g/ha) showed lower season-long male mealybug trap captures in the MD treatments than in the control (χ^2^ = 16.664, df = 2, *p* < 0.001), with no significant effect of sample date (χ^2^ = 2.379, df = 1, *p* = 0.124), but there was a treatment × date interaction (χ^2^ = 10.429, df = 2, *p* < 0.001) (Figure 2A). Pairwise comparisons showed fewer captures at both MD rates than the control, and no difference between MD treatments. Trap captures were not completely ‘shut down’ in the MD treatments and the lower dose (20 g/ha) showed a sharp increase from September to October, whereas trap counts with the higher (60 g/ha) rate remained low until November (Figure 2A). The mid-October drop in trap captures can be attributed to the post-harvest application of chlorpyrifos in 2 of the 4 vineyards. There was a treatment effect on fruit cluster ratings (χ^2^ = 23.314, df = 6, *p* < 0.001), with lower damage in the Isomate 20 g/ha plots; overall, *Pl. ficus* damage was very high, from 32–47% of the clusters rated (Figure 2B).

The second 2004 trial compared two Isomate loads (4.1 and 8.3 g/ha), CheckMate Flowable (32 g/ha), and CheckMate dispenser (33.3 g/ha). For trap captures, the Isomate and CheckMate materials were compared separately to the control. The two Isomate rates showed no treatment effects on season-long male captures (χ^2^ = 0.115, df = 2, *p* = 0.891) with no effect of sample date (χ^2^ = 0.357, df = 1, *p* = 0.551) or treatment × date interaction (χ^2^ = 0.312, df = 2, *p* = 0.732) (Figure 3A). Season-long captures of males with both the CheckMate Flowable and dispensers were lower than the control (χ^2^ = 3.547, df = 2, *p* = 0.029) with a sample date effect (χ^2^ = 11.499, df = 1, *p* < 0.001) but no treatment × date interaction (χ^2^ = 1.322, df = 2, *p* = 0.267) (Figure 3B). Pairwise comparisons showed both the flowable and dispenser treatments were lower than the control but were not different from each other. Harvest rating of fruit clusters showed a significant treatment effect (χ^2^ = 63.94, df = 12, *p* < 0.001) and pairwise comparisons showed all blocks treated with MD formulations had less damage than the control, with no separation among MD treatments (Figure 3C). As in the Isomate trial, *Pl. ficus* densities and damage were high as indicated by mealybug infestation levels of 16–33% of the rated clusters.

In 2005, the same treatment plots were used to compare the Isomate (8.3 g a.i./ha), CheckMate Flowable (32 g/ha), and CheckMate dispensers (33.3 g/ha). There was a season-long effect on male trap captures (χ^2^ = 6.740, df = 3, *p* < 0.001) with an effect of sample date (χ^2^ = 14.750, df = 1, *p* < 0.001) and a treatment × date interaction (χ^2^ = 2.772 df = 3, *p* = 0.040) (Figure 4A). Pairwise comparisons showed both the CheckMate Flowable and dispenser treatments were lower than the control and the Isomate at 4.1 g/ha but were not different from each other. Harvest rating of fruit clusters across all treatments showed a treatment effect (χ^2^ = 21.123, df = 9, *p* = 0.012) and pairwise comparisons showed separation of only the Isomate from the control and CheckMate dispenser treatments (Figure 4B). As in the 2004 trials, *Pl. ficus* densities and damage were high as indicated by mealybug infestation levels of 7.2–11% of the rated clusters, although they were lower than in 2004.

The five minute counts largely supported pheromone trap captures and cluster sample results. In 2004, there was a season-long treatment effect on total mealybugs per vine (χ^2^ = 3.540, df = 4, *p* = 0.007) with a sample date effect (χ^2^ = 108.71, df = 1, *p* < 0.001) but no treatment × date interaction (χ^2^ = 2.11, df = 2, *p* = 0.077). Pairwise comparisons showed that mealybug counts in plots treated with CheckMate Flowable and dispensers were significantly lower than the control, and the flowable formulation was also lower than the Isomate 8.3 g/ha treatment. Across all sample dates, there was a treatment effect on total mealybug densities (χ^2^ = 5.165, df = 4, *p* < 0.001), with pairwise comparison showing Isomate (8.3 g/ha), CheckMate Flowable, and CheckMate dispensers lower than the control (Figure 5A). Whereas there was a treatment effect on density, there was no discernable pattern on *Pl. ficus* development stages present (Figure 5A). For example, there was a treatment effect on the number of ovisacs (χ^2^ = 3.462, df = 4, *p* = 0.008) and first instars (χ^2^ = 3.497, df = 4, *p* = 0.007), with pairwise comparisons showing less of each stage in the CheckMate Flowable and CheckMate dispensers than in the control, and fewer ovisacs in the CheckMate treatments. However, this was related to overall density and when analyzing the ratio of different developmental stages there was no treatment × developmental stage interaction (χ^2^ = 1.083, df = 16, *p* = 0.365) on the ratio of the different stages among treatments. Treatment and location affected total mealybugs on the vine (χ^2^ = 20.168, df = 5, *p* < 0.001) and a location × treatment interaction (χ^2^ = 5.192, df = 20, *p* < 0.001), with more *Pl. ficus* on the canes, leaves, and fruit in the control and Isomate (4.1 g/ha) than the CheckMate Flowable (32 g/ha) and dispenser (33.3 g/ha) treatments (Figure 5B).

The 2005 data supported the 2004 trial; the three minute counts showed a season-long treatment effect on total mealybugs per vine (χ^2^ = 5.206, df = 3, *p* < 0.001), as well as an effect of sample date (χ^2^ = 4.799, df = 1, *p* = 0.029) and a sample treatment × date interaction (χ^2^ = 3.380, df = 3, *p* = 0.018). Pairwise comparisons showed the Isomate, CheckMate Flowable, and CheckMate dispensers were significantly lower than the control, and the flowable was lower than the Isomate treatment (averages over all sample dates were control: 7.8 ± 0.6, Isomate: 5.6 ± 0.5, CheckMate Flowable: 4.2 ± 0.5, and CheckMate dispenser 4.9 ± 0.5). Developmental stage structure appeared similar among treatments (Figure 6A), but when comparing the ratio of stages present over the season, there was a significant treatment × stage interaction (χ^2^ = 2.696, df = 12, *p* = 0.001). Individual pairwise comparisons by stage showed a treatment effect for only the percentage of ovisacs present, with more ovisacs in the flowable than the CheckMate dispenser treatments. There was an effect of the location on total mealybugs on the vine (χ^2^ = 3.309, df = 5, *p* = 0.006) but no location × treatment interaction (Figure 6B); pairwise comparisons showed more *Pl. ficus* found on the ground, trunk, arm, cane, and leaves than the fruit (season-long analyses).

In 2004, 4350 adult *Pl. ficus* were collected and isolated in gelatin capsules (range 859–888 per treatment). From these, 41.6% produced eggs, with 48.7 ± 1.7, 38.9 ± 1.7, 39.1 ± 1.6, 38.0 ± 1.6 and 39.1 ± 1.6% of the females producing eggs in the control, Isomate 4.1 g/ha, Isomate 8.3 g/ha, flowable, and CheckMate dispensers, respectively (χ^2^ = 7.068, df = 4, *p* < 0.001, with all MD treatments lower than the control). Of those producing eggs, production ranged from 3–458 eggs per female, with an unexpected treatment effect (χ^2^ = 2.984, df = 4, *p* = 0.018), with fewer eggs per female in the control (49.3 ± 2.2 eggs per female) than the Isomate 8.3 g/ha or CheckMate Flowable (59.5 ± 2.8 and 59.1 ± 2.6 eggs per female, respectively). All parasitoids reared were *A. pseudococci*; parasitism ranged from a low in the control of 7.1 ± 1.6% to a high in the CheckMate Flowable of 11.8 ± 2.0%, but there was no treatment effect (χ^2^ = 1.216, df = 4, *p* = 0.302).

In 2005, 1855 adults were collected and isolated (range 455–470 per treatment). From these, 37.3% produced eggs, with 39.8 ± 2.3, 38.7 ± 2.2, 38.3 ± 2.1, and 32.1 ± 2.0% of the females producing eggs in the control, Isomate 4.1 g/ha, CheckMate Flowable and CheckMate dispensers, respectively (χ^2^ = 2.391, df = 3, *p* = 0.067). Of those producing eggs, production ranged from 2–412 eggs per female with fewer eggs per female in the Isomate (79.4 ± 4.3) than with the CheckMate dispenser (93.9 ± 4.4) (χ^2^ = 2.998, df = 3, *p* = 0.030).

The leaf samples sought to determine if initial *Pl. ficus* density per vine would affect pheromone effectiveness. The mealybug population is largely on the trunk and canes during the spring and early summer [2], and this pattern was followed in both 2004 and 2005, with few mealybugs recorded on leaves before June, and an increase in *Pl. ficus* density thereafter. In 2004, before mid-July, there was no treatment difference in *Pl. ficus* per leaf, but there was, as planned, a difference in low, medium, and high infestation levels across all treatments (χ^2^ = 3.798, df = 2, *p* = 0.023) but no infestation × treatment interaction. During the July–August period, with higher *Pl. ficus* per leaf densities, there was a significant treatment effect on overall *Pl. ficus* density (χ^2^ = 3.675, df = 4, *p* = 0.006) and a difference among infestation levels (χ^2^ = 36.301, df = 2, *p* < 0.001), but still no treatment × infestation interaction. Pairwise comparisons showed Isomate (8.3 g/ha) and CheckMate Flowable and dispensers were significantly lower than the control (Figure 7A).

A similar pattern followed in 2005; there was no treatment difference in *Pl. ficus* per leaf, but with lower overall *Pl. ficus* densities, there was also no initial difference measured among low, medium, and high infestation levels across all treatments and no infestation × treatment interaction. After July, there was a treatment effect (χ^2^ = 2.782, df = 4, *p* = 0.028) with pairwise comparisons showing more *Pl. ficus* in the control than Flowable treatment (Figure 6B). There was also a difference among infestation levels (χ^2^ = 5.172, df = 2, *p* = 0.006), with more mealybugs in the high than low initial densities, but there was no treatment × infestation level interaction. The density patterns in both 2004 and 2005 show only the flowable formulation brought the medium density population closer to the low density, and the higher CheckMate rates had the low population near zero (Figure 7A,B).

### 3.2. Field Trials—2006 and 2007

The 2006 and 2007 trials used larger plots, and a single and more consistent insecticide application (systemic imidacloprid or foliar buprofezin). In both years, CheckMate dispensers were applied at a rate of 61.6 g/ha. In 2006, season-long captures of males were lower in the MD treatment than in the control (χ^2^ = 8.788, df = 1, *p* = 0.003) with a sample date effect (χ^2^ = 5.798, df = 1, *p* = 0.016) but no treatment × date interaction (χ^2^ = 0.3.445, df = 1, *p* = 0.064) (Figure 8A). In 2007, season-long captures of males were weakly lower in the MD treatment than in the control (χ^2^ = 4.087, df = 1, *p* = 0.044) with a strong sample date effect (χ^2^ = 124.04, df = 1, *p* < 0.001) and no treatment × date interaction (χ^2^ = 1.281, df = 1, *p* = 0.259) (Figure 8B).

Harvest rating of fruit clusters showed no treatment effect in 2006 (χ^2^ = 6.018, df = 3, *p* = 0.111) with relatively high damage levels in both treatments, with clusters rated as damaged (2) or unmarketable (3) comprising 6.8 and 3.1% in the control and 6.2 and 1.7% in the MD treatment, respectively (Figure 9A). In 2007, there was lower cluster damage in the MD treatment (χ^2^ = 18.313, df = 3, *p* < 0.001), but damage levels were still unacceptably high with clusters rated as damaged (2) or unmarketable (3) comprising 12.4 and 7.0% in the control and 9.6 and 3.6% in the MD treatment (Figure 9B).

In 2006, the presence–absence sampling showed 45.7 ± 0.7% of the vines were infested across all vineyard blocks (range 23.3 ± 1.5 to 67.7 ± 1.7%). There was little difference in infestation levels between plots treated with imidacloprid (50.5 ± 0.1%) and buprofezin (47.6 ± 0.1%), and for that reason, data were combined. The presence–absence sampling allowed more vines to be searched (*n* = 4795); however, it provided a poor comparison of mealybug density and showed no differences among treatments (χ^2^ = 0.018, df = 1, *p* = 0.893) with a sample date effect (χ^2^ = 102.81, df = 1, *p* < 0.001) but no treatment × date interaction (χ^2^ = 0.243, df = 1, *p* = 0.622). In 2007, an adjustment was made in the sampling program, by rating vines using the categorical rating system. 

There was no treatment difference in May, just prior to the deployment of MD plastic dispensers, or June, as mealybug density began to increase. There were more mealybugs in the control than in the MD treatment in July and August, based on categorical rankings (Figure 10). The difference was economically important because nearly 18% more vines had the two highest categorical rankings in the control treatment. The newly developed categorical ranking of mealybug density allowed more vines to be searched; a total of 3900 vines was sampled over the four sample dates.

In 2007, all vineyards within a 450 ha area surrounding four of six MD treatments were monitored during the peak flight period (Figure 11). Density patterns (*Pl. ficus* per week per trap) showed mealybug ‘hot spots’ with respect to the location of MD plots. With one exception (October, northwest plot), MD sites had low *Pl. ficus* trap captures. Moreover, no ‘halo’ effect was evident, suggesting that MD was not pulling in males from nearby vineyards.

### 3.3. Dispenser Load and Release Rate

Optimizing the rate of release of the synthetic pheromone from dispensers is a critical step to building successful mating disruption programs. Release of the pheromone is based on membrane permeability (Figure 1D; black dot in center of dispenser is the permeable area), which can be adjusted through a range of manufacturing processes. Temperature also affects release rates whereby higher temperatures result in a faster release. In 2004 trials, the pheromone was released far too quickly, and the dispensers were depleted in 4–5 weeks after field deployment, regardless of the load (60 vs. 100 mg) or location (Parlier, CA vs. the Coachella valley, CA) (Figure 12A). Based on these results, our commercial partner (Suterra, LLC.) altered the permeability of the membrane. In 2005 and 2006, the initial release was near target rate, but by mid-August—a critical period for mating—the release rate slowed such that by the end of the season, only 40% of the synthetic pheromone had been released in the Parlier, CA trial (Figure 12B). Following additional changes to the membrane, the 2007 trial achieved a more consistent release rate from May to September, with a slight reduction in the release rate in cooler October, and about 80% of the pheromone released throughout the season.

### 3.4. Pheromone Purity

During the season, percentage infested vines ranged from 30.1 ± 3.9 to 54.9 ± 4.3%, but there was no treatment effect in July (χ^2^ = 20.65, df = 3, *p* < 0.001). Harvest cluster evaluation showed that all MD treatments had lower damage than the control, but there was no difference between 95 and 99% purity or between 99% purity production batches (χ^2^ = 50.211, df = 9, *p* < 0.001; pairwise comparisons set at α < 0.0083).

## 4. Discussion

Research reported herein documents initial MD trials for *Pl. ficus* in California that led to the registration and commercialization of MD programs for a mealybug pest. Trials of MD as a control strategy for *Pl. ficus* were first reported using a microencapsulated formulation [49], but trials using plastic and rope dispensers were carried out concurrently. Researchers in Europe, South America, and Israel have also tested various types of passive dispensers for *Pl. ficus* MD [50,51,53,54,56]. Although MD programs are now successfully being used for *Pl. ficus*, the research presented here has relevance to improvements to these ongoing programs as well as highlighting needed research directions. In the California studies reported herein, two dispenser types were used, CheckMate plastic pouch dispensers (Suterra LLC.) and Isomate plastic tube dispensers (ShinEtsu Inc. and Scentry Inc.). However, a direct comparison of dispenser type was not made because the pheromone load rate varied from 7.5 to 100 mg per dispenser. A relatively consistent deployment pattern of every other vine and every row resulted in pheromone application rates from 4.15 to 61.7 g (AI)/ha. Across all trials during this period, we report either a treatment effect on pheromone trap captures or crop damage, regardless of the pheromone dispenser type or load (Figure 2, Figure 3, Figure 4, Figure 8 and Figure 9). It was also evident that near zero levels of trap capture were never achieved (Figure 2A, Figure 3A,B, Figure 4A and Figure 8). From this work, it became evident that seasonal coverage was as or more important than the total amount of pheromone deployed.

Another factor that became evident was that *Pl. ficus* densities in experimental plots were unacceptably high for commercial vineyards, with cluster damage often above 10% (Figure 2B, Figure 3C, Figure 4B). As discussed by Walton et al. [49] for *Pl. ficus* and by Witzgall et al. [57] for MD programs in general, pheromones are increasingly efficient at lower pest population densities. The use of vineyards with relatively high *Pl. ficus* populations was done deliberately, to attempt to show an economic effect on cluster infestation, without having to sample thousands of fruit clusters per plot to show significant differences between treatments and controls. For the amount of crop damage recorded, *Pl. ficus* trap captures were surprisingly low. We suggest this is due, in part, to the post-harvest insecticide applications used in the most heavily infested blocks that would lower adult male *Pl. ficus* captures in September and October, a period of peak flight activity, and to the biased female: male ratio that we have observed in California vineyards from May to August. Another possibility is a population of *Pl. ficus* that did not respond to the pheromone; Kol-Maimon et al. [58] showed different *Pl. ficus* pherotypes, as defined according to the male behavior to pheromone compounds, with some responding to different pheromone components and having different flight performance, both in relationship the population density. Another contributing factor may have been the relatively small doses used in the pheromone lures (60–100 mg per lure), which may have been outcompeted by females in high density *Pl. ficus* populations, exacerbated by pheromone blowing over from treated blocks into control blocks; moreover, in these initial trials the plot sizes were relatively small, placing treated and control blocks close together.

From 2004 to 2007, release rates for the CheckMate dispensers were less than satisfactory, with pheromone depleted either too quickly (2004) or not during the growing season (2005 and 2006) (Figure 12). Given that only 40% of the pheromone had been released in the early studies, the actual dose was lowered by the same percentage in the 2005 and 2006 studies. The flowable formulation at 32 g/ha (Figure 3B) and 16 g/ha (Figure 4A) performed as well as or better than the dispensers at a comparable application rate (33.3 g/ha). For almost all trials, there was a resurgence of *Pl. ficus* trap captures in mid-July or August at which point a post-harvest insecticide spray was often applied. Thus, season-long coverage or late-season coverage appears to be as or more important than dose, at least for the doses tested in this study. Developing a dispenser that provides season-long pheromone emission and determining the optimal number of dispersers to deploy per hectare may be a future goal.

Seasonal coverage is contingent on the dispenser release rate or the number of flowable applications made. Dose per hectare is a different measurement and, for dispensers, is based largely on the deployment density and load, whereas for flowable formulations dose is based on the initial application rate and the number of applications per season. For dispensers, ambient temperature affects pheromone release rate, which in turn affects seasonal coverage. The CheckMate dispenser produced and tested in 2007 (Figure 12B) is similar to the product which is now commercially available and, indeed, researchers report field efficacies of 4–5 months, whereas rope (Isomate) dispensers provide slightly longer coverage [50,54]. A common misconception is that increasing the number of dispensers deployed or the application rate of flowable material will extend the field lifetime of the pheromone. In warmer vineyard production regions, such as California’s Central Valley, to extend coverage later in the season, dispensers should be deployed later. A flowable formulation has the advantage of being more easily deployed to target *Pl. ficus* during any seasonal period; however, labor and equipment constraints around harvest often limit later season applications. Dispenser loads in these trials varied from 4.15–100 mg and the deployment rates were consistently near or above 600 per ha. Current commercial products typically have higher loads (150–180 mg per dispenser), but most other trials use deployment rates near or above 600 dispensers/ha [50,51,54,59]. An informative study in Italy tested deployment rates of 300, 400, and 500 dispensers/ha (180 mg per dispenser) and showed that all treatments had a reduction in *Pl. ficus* cluster damage compared with untreated controls [53], adding more support to the hypothesis that, above a certain threshold, season-long coverage may be as important as dose in *Pl. ficus* MD.

In the 2004–2007 trials, treatment effect on population biology and structure was assessed, in addition to *Pl. ficus* density and damage. Surprisingly, there was not a clear difference in population age structure, even when there was a significant reduction in *Pl. ficus* density (Figure 5A and Figure 6A). Our initial hypothesis was that, as a percentage of the population, fewer ovisacs would be found in plots treated with mating disruption. However, the large number of annual generations, overlapping generations and production of non-viable ovisacs from unmated females may have obscured any differences in age structure. A greater difference in egg production was expected; however, while there were fewer eggs per female in all MD treatments in 2004, the difference was only about 20%. For this study, 100–200 ovisacs were collected from each plot, and there were fewer viable ovisacs but similar numbers of eggs in MD plots. These results agree with Waterworth et al. [60], who showed that *Pl. ficus* females must mate to produce viable eggs, but that multiple mating did not increase reproductive output. There was also not a clear treatment effect on *Pl. ficus* distribution on the vine (Figure 5B and Figure 6B). Mealybugs likely follow the movement of essential nutrients in the phloem and progress from the trunk and cordon to the leaves and then to the fruit as the season progresses [2]. There was an expectation that fewer mealybugs would be on the leaves and fruit, as a percentage of the population, due in part to higher parasitism rates by *A. pseudococci*. Franco et al. [61] found a kairomonal response of *A*. nr. sp. *pseudococci* to *Pl. ficus* and *Pl. citri* pheromones, and it was assumed that MD could lead to higher percentage parasitism, but overall parasitism in this study was generally low.

Our trials also evaluated two cost-saving measures to optimize mating disruption technology for *Pl. ficus*. First, a comparison of synthetic pheromone purity showed no difference among two batches of 99% and one batch at 95% chemically pure racemic lavandulyl senecioate. Increases in purity require more rigorous distillation processes [42], which increases the costs of production that in turn are passed on to the consumer. This work helped to define the production practices required to produce an effective product at an affordable price. While these trials were being conducted, mating disruption devices that released timed bursts of pheromone (e.g., Puffer technology) were being developed for lepidopteran pests in orchards e.g., [62], and an advantage of Puffer technology is that pheromone release can be programmed for different periods of the day. A disadvantage of passive (plastic pouch or rope dispensers or flowables) is that they release continuously, and more pheromone is emitted during warmer periods of the day, when *Pl. ficus* flight activity is relatively low. Aerosol delivery systems such as Puffer devices have advantages over passive dispensers, as they are cheaper to apply because they are applied at much lower densities (<5 per ha), and can release pheromones at selected time intervals when the target pest is active [63]. However, aerosol devices drastically reduce the number of point sources per ha, which could reduce overall MD effectiveness [64], especially for vineyard mealybugs that tend to have clumped distributions [40]. More work is needed to follow up on the dispenser density trials conducted by Lucchi et al. [53], which suggest deployment rates nearly half of that used in this study can still provide *Pl. ficus* suppression. Captures of mealybugs in pheromone traps placed both inside and outside of the MD plots (Figure 11) would suggest that dispensers had little effect on *Pl. ficus* trap captures just 0.2 km away.

## 5. Conclusions

In most trials, MD treatments, regardless of the applied dose, impacted either trap capture of adult male *Pl. ficus* or crop damage, but the effect is best described as varying levels of suppression rather than economically acceptable control. A critical factor was the rapid (2004) or incomplete (2005–2006) release of the pheromone from the CheckMate dispensers used in most trials. The successful 2007 batch of dispensers set the prototype for the first commercial product, and since then, there have been advances in plastic pouch and rope dispensers as well as microencapsulated flowable formulations. Maintaining season-long pheromone emission from dispensers at effective rates is an ongoing issue. In California, vineyard managers prefer early dispenser deployment for practical reasons, as labor crews are more available in March–April than later in the season, and deployment is easier before the vine cane growth covers the trellis wire. The disadvantage of early deployment is that dispensers often will not last until the critical August–October period when *Pl. ficus* flight activity is highest, leading to post-harvest insecticide applications to augment MD. Furthermore, incomplete MD coverage during the early flight period, particularly following a warm winter, could result in late-season population explosions that are not adequately controlled by MD and require further intervention. Seasonal coverage warrants further study to identify critical periods and optimize coverage strategies, both in cooler (coastal) and warmer (interior) regions of California. Dispenser deployment per hectare or the effect of different numbers of point sources also need to be further studied as dispensers are manually hung on the trellis wire, and any reduction in dispenser deployment that does not concurrently reduce effectiveness decreases both material and labor costs.

## Figures and Tables

**Figure 1 insects-11-00635-f001:**
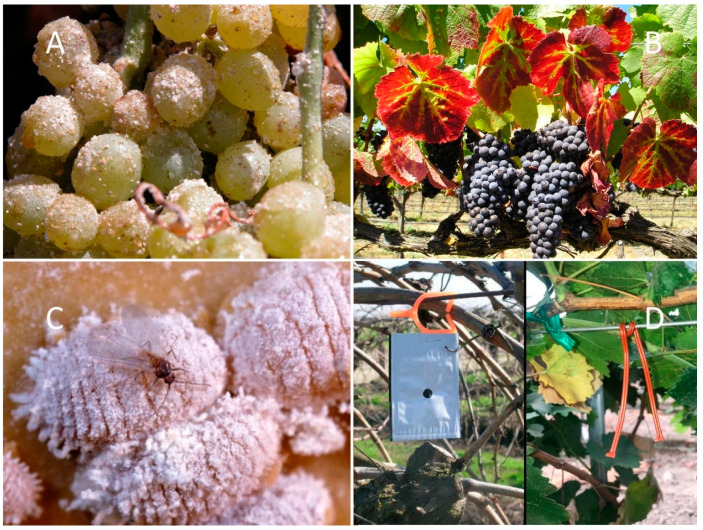
*Planococcus ficus* is one of the more important vineyard pests globally; shown here (**A**) damage to a table grape cluster from excessively high population density, (**B**) the reddened leaves symptomatic of grape leafroll disease (GLD) on a red-cultivar wine grape caused by grape leafroll-associated viruses transmitted by mealybugs, (**C**) mature female and an adult male *Pl. ficus*. and (**D**) the CheckMate dispenser (left) and Isomate rope dispensers (right) used in the studies described herein (note that commercial dispensers have gone through improvements since this work).

**Figure 2 insects-11-00635-f002:**
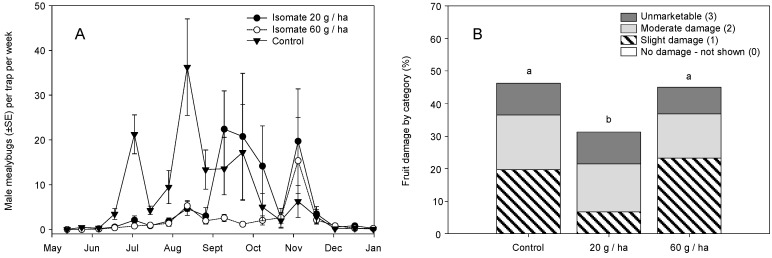
2004 field studies showing (**A**) season-long adult male *Pl. ficus* captures (mean ± SEM) in pheromone-baited traps in plots with Isomate dispensers (ShinEtsu) hung at rates of 20 and 60 g (AI) per ha and (**B**) categorical ranking of mealybug damage to fruit clusters at harvest time (different letters above each bar represent significant differences, pairwise comparisons using an experiment-wide error rate at *p* < 0.016).

**Figure 3 insects-11-00635-f003:**
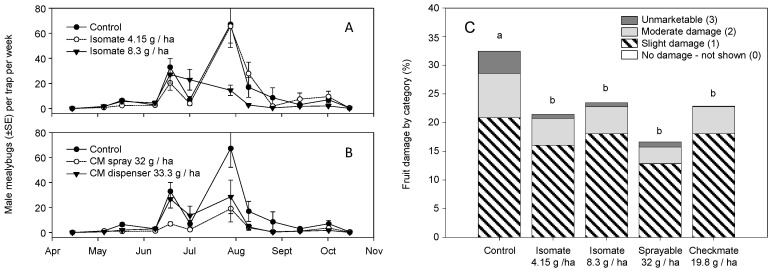
2004 field studies of season-long adult male *Pl. ficus* (mean ± SEM) captures in pheromone-baited traps in plots with (**A**) Isomate dispensers (Scentry Inc.) hung at rates of 4.15 and 8.3 g a.i./ha; (**B**) CheckMate Flowable applied four times for a rate of 33.3 h a.i./ha and CheckMate dispensers hung at 19.8 g a.i./ha; (**C**) the categorical ranking of mealybug damage to fruit clusters at harvest time across all treatments (different letters above each bar represent significant differences, pairwise comparisons using an experiment-wide error rate at *p* < 0.005).

**Figure 4 insects-11-00635-f004:**
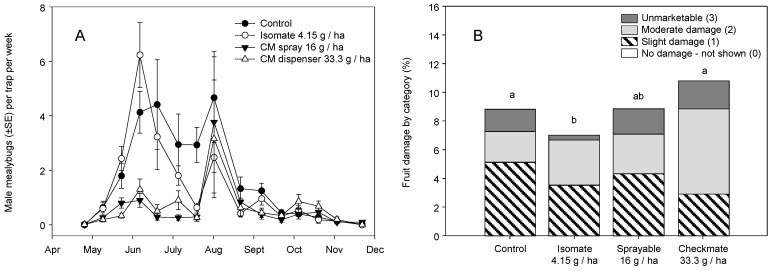
2005 field studies of season-long adult male *Pl. ficus* (mean ± SEM) captures in pheromone-baited traps in plots with (**A**) Isomate dispensers (Scentry Inc.) hung at rates of 4.15 g a.i./ha and CheckMate Flowable formulation applied two times for a rate of 16 h a.i./ha and CheckMate dispensers hung at 19.8 g a.i./ha; (**B**) the categorical ranking of mealybug damage to fruit clusters at harvest time across all treatments (different letters above each bar represent significant differences, pairwise comparisons using an experiment-wide error rate of *p* < 0.01.).

**Figure 5 insects-11-00635-f005:**
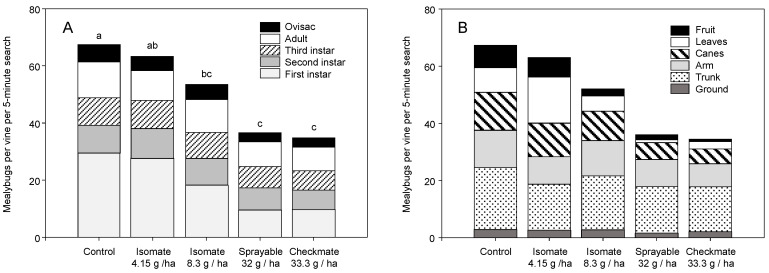
2004 field studies of five minute per vine searches for *Pl. ficus* in plots treated with no mating disruption (MD), Isomate (4.15 g/ha and 8.3 g/ha), CheckMate Flowable (32 g/ha), or CheckMate dispensers (33.3 g/ha) showing (**A**) season-long average *Pl. ficus* by developmental stage category and (**B**) *Pl. ficus* by location on the vine. Different letters above each bar represent significant differences in the total number of mealybugs in each treatment, *p* < 0.05.

**Figure 6 insects-11-00635-f006:**
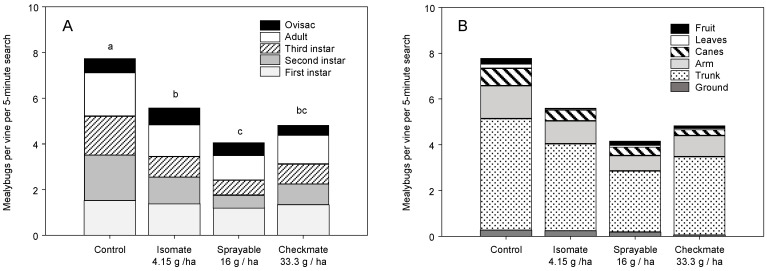
2005 field studies of five minute per vine searches for *P. ficus* in plots treated with no MD, Isomate (4.15 g/ha), CheckMate Flowable (16 g/ha), or CheckMate dispensers (33.3 g/ha) showing (**A**) season-long average *Pl. ficus* by developmental stage category and (**B**) *Pl. ficus* by location on the vine. Different letters above each bar represent significant differences of the total number of mealybugs in each treatment, *p* < 0.05.

**Figure 7 insects-11-00635-f007:**
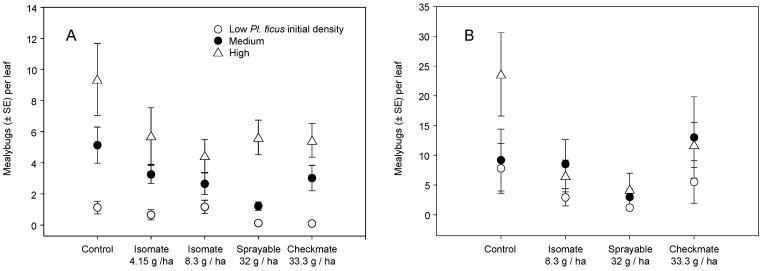
Vines were categorized as having an initial (April-May) low, medium, or high *Pl. ficus* population density and then compared for relative increases in (**A**) 2004 with plots treated with no MD, Isomate (4.15 g/ha and 8.3 g/ha), CheckMate Flowable (32 g/ha), or CheckMate dispensers (33.3 g/ha) and (**B**) 2005 with plots treated with no MD, Isomate (4.15 g/ha), CheckMate Flowable (24 g/ha), or CheckMate dispensers (33.3 g/ha).

**Figure 8 insects-11-00635-f008:**
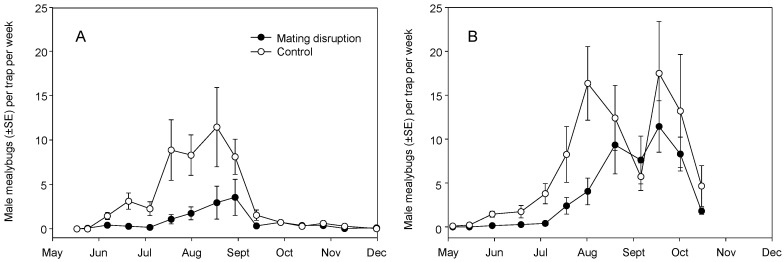
Season-long adult male *Pl. ficus* (mean ± SEM) captures in pheromone-baited traps in plots with CheckMate dispensers hung at 61.5 g/ha were lower than the control in (**A**) 2006 and (**B**) 2007.

**Figure 9 insects-11-00635-f009:**
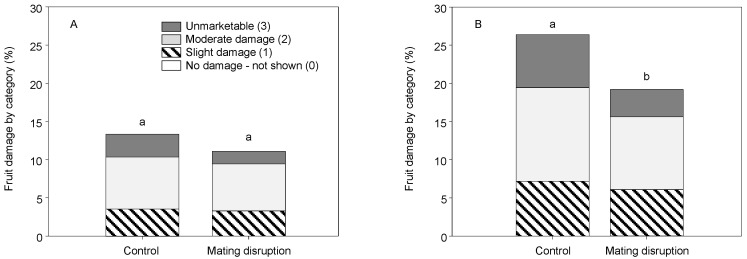
Categorical ranking of mealybug damage to fruit clusters at harvest time across captures in plots with CheckMate dispensers hung at 61.5 g/ha were not different from the control in (**A**) 2006 and lower than the control in (**B**) 2007 (different letters above each bar represent significant differences).

**Figure 10 insects-11-00635-f010:**
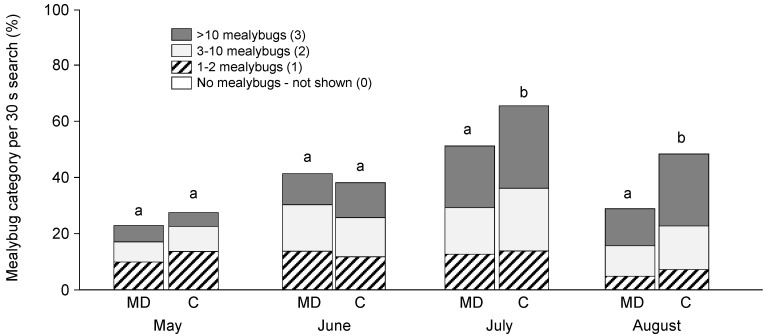
The categorical ranking of mealybug density showed no treatment difference in 2007 in May (χ^2^ = 4.639, df = 3, *p* = 0.200) or June (χ^2^ = 2.854, df = 3, *p* = 0.415) and significantly less damage on vines in the mating disruption (MD) than the control (C) in July (χ^2^ = 20.65, df = 3, *p* < 0.001) and August (χ^2^ = 41.05, df = 3, *p* < 0.001). For each month, different letters above each bar represent significant differences.

**Figure 11 insects-11-00635-f011:**
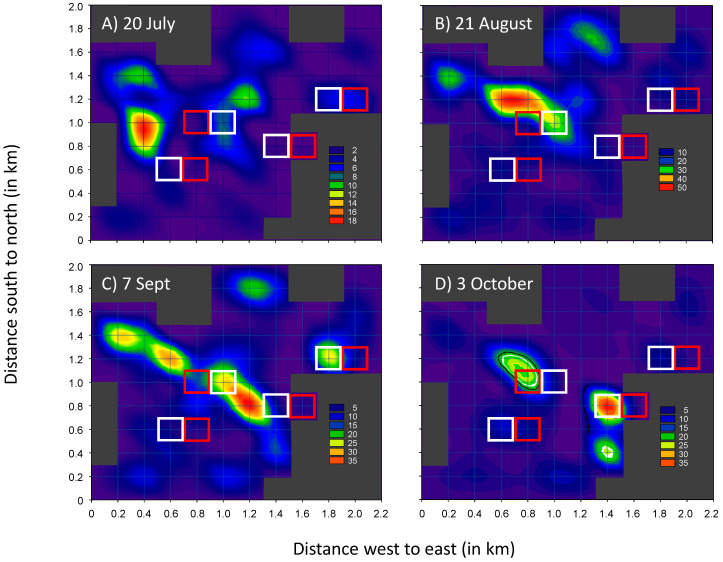
Adult male *Pl. ficus* trap captures in a 450 ha region surrounding MD plots (red squares) and the corresponding control plots (white squares) that received CheckMate dispensers hung at 61.5 g/ha in 2007. For each graph, key in lower right is *Pl. ficus* density per week for the different monthly sampling periods (**A**–**D**). Dark gray areas indicate non-grape crops or buildings. Maps were constructed using the Contour Graph function in SigmaPlot for Windows Version 12.5, Systat Software, Inc.

**Figure 12 insects-11-00635-f012:**
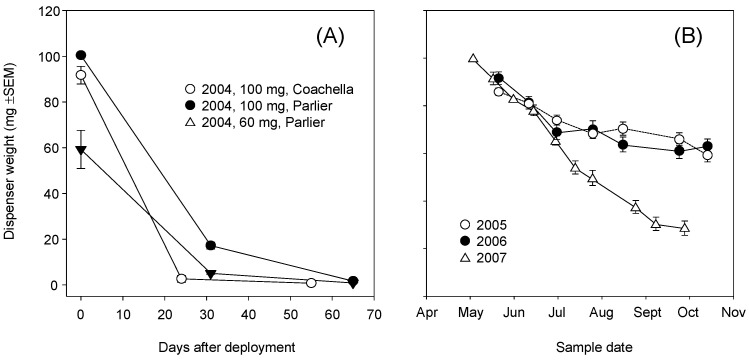
Release of synthetic pheromone for (**A**) 2004 and (**B**) 2005–2007, as determined by the weight of pheromone remaining in dispensers placed in the field (Parlier or Coachella valley) and collected every 2 weeks.

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
