# Peer review of "Development of a Mating Disruption Program for a Mealybug, Planococcus ficus, in Vineyards"

_insects, 2020, doi:10.3390/insects11090635_

Round 1

Reviewer 1 Report

This is a quite valuable report that shows a “historical record” for the initial development of a mating disruptant to control Planococcus ficus, a serious pest in vineyards in many regions. The results seem unclear and vary among seasons, at least partly because of differences of versions of mating disruptants used in field experiments. But, overall, the authors found an effect on trap captures of adult males or crop damages. As the authors concluded, the effect is not perfect to control this mealybug under an economically acceptable level, but the mating disruptant would be offered as one of alternative tactics in IPM programs. I think this paper gives many hints to a broad range of applied entomologists.

I would like to give only a question about the attractiveness of the pheromone chemical, lavandulyl senecioate. I guess mealybug densities in the experimental sites were considerably high, but pheromone trap captures were ~60/trap/week at maximum. Some previous reports (e.g. Kol-Maimon et al. doi: 10.1007/s00114-010-0726-3) show there are multiple pherotypes in P. ficus. Is a single lavandulyl senecioate enough to attract males? Is it possible that some unknown minor component is present in the pheromone system of this species and may improve the limited efficacy of the mating disruptant? Please give some descriptions about this point in the manuscript, if possible.

Author Response

As always, the Reviewers’ comments greatly improved the manuscript. We have accepted all changes, as described below in more detail, and we have carefully looked over the manuscript and made additional minor corrections to keep terminology consistent within the manuscript and correct awkward sentences. Below are more detailed responses to each of the Reviewer’s comments. Here are our responses to all Reviewers

Reviewer 1.

Reviewer was very complimentary and this is appreciated. There was one primary concern, “I would like to give only a question about the attractiveness of the pheromone chemical, lavandulyl senecioate. I guess mealybug densities in the experimental sites were considerably high, but pheromone trap captures were ~60/trap/week at maximum. Some previous reports (e.g. Kol-Maimon et al. doi: 10.1007/s00114-010-0726-3) show there are multiple pherotypes in P. ficus. Is a single lavandulyl senecioate enough to attract males? Is it possible that some unknown minor component is present in the pheromone system of this species and may improve the limited efficacy of the mating disruptant? Please give some descriptions about this point in the manuscript, if possible.

The pheromone used was the original formulation developed from Pl. ficus in California where Dr. Millar’s lab found and identified lavandulyl senecioate and the corresponding alcohol, lavandulol and then tested each compound alone, and the blend of the two, and found that lavandulol had no activity alone, and the blend of lavandulol and lavandulyl senecioate was not significantly different than lavandulyl senecioate alone. So, there was no indication that the pheromone might be a blend.

Still, we have read the Kol-Maimon manuscript and cannot be certain that there are not multiple strains of Pl. ficus in California. We do note, however, that as mentioned in the article, many of the vineyards received a post harvest application of chlorpyrifos, because of the high Pl. ficus densities, and this would have certainly lowered the Pl. ficus trap captures in September and October when their numbers are highest. We have also observed low trap captures when Pl. ficus densities are very high, and assume that this is related to the ration of the females: males produced (which can vary). We have cited the Koi-Maimon article and commented on this possibility in the Discussion as follows:

For the amount of crop damage recorded, Pl. ficus trap captures were surprisingly low. We suggest this might have been due, in part, to the post-harvest insecticide applications used in the most heavily infested blocks that would lower adult male Pl. ficus captures in September and October, a period of peak flight activity, and to the biased female: male ratio that we have observed in California vineyards from May to August. Another possibility is a population of Pl. ficus that did not respond to the pheromone; Kol-Maimon et al. [64] showed different Pl. ficus pherotypes, as defined according to the male behavior to pheromone compounds, with some responding to different pheromone components and having different flight performance, both in relationship the population density.

Reviewer 2

Reviewer 2 was also quite complimentary and made no directed suggestions for improvements but did check the boxes suggesting that we should double check the spelling and that the M&M could be better described.

We have checked the spelling and looked again at the M&M. We do agree that more details could have been added in describing collection methods, but we do cite Walton et al. 2006 for greater details. Because of the length of the manuscript, we decided to keep the M&M description to the minimum possible, while still letting the reader know our procedures. It is our preference not to add more detail.

Reviewer 3

This Reviewer had the most comments, and we have included all of his suggestions, as follows:

1) Keywords have now been placed in alphabetical order

2) Caption for Figure 1 should have scientific names in italics. This is strange as in my submitted copy Planococcus ficus is in italics… but we have checked and confirmed this in the revision.

3) Reviewer requests that we provide order and family for Lobesia botrana. We have done this, but note that in most journals when citing another article it is not needed to also provide Order and Family for the insect.

4) A request was made to provide more information on the cited article about the sex pheromone for Pl .citri. We have changed this sections to “[36,37]. It is known that mature female Pl. citri emit a sex pheromone to attract the winged adult males [38], and this pheromone, initially identified as (1-)-(+)-2,2-dimethyl-3-(1-methylethenyl)cyclobutanemethanol acetate, can be synthesized [39]. With this information as a starting point, a semiochemical approach for Pl. ficus was initially focused on developing a monitoring tool for this invasive pest in California…”

5) A request was made to provide more information on the herbicide (Manufacturer, City, Country “The vineyards were ~15 year-old Thompson Seedless cv., with vines trained to a single- or double-T-trellis and a clean vineyard floor maintained by running a disc over the row middles and applying an herbicide (glyphosate) to the berms” As we used the generic chemical name for “Roundup” we felt that we did not need to add the product name, but we have complied and changed this to, “The vineyards were ~15 year-old Thompson Seedless cv., with vines trained to a single- or double-T-trellis and a clean vineyard floor maintained by running a disc over the row middles and applying the herbicide Roundup® (Monsanto Co., Creve Coeur, MO., USA) to the berms.”

6) Similarly, the reviewer wanted mention of dispensers per acre removed, and the trade names for insecticide applied and so we have done this changing,

There was a 20 row buffer between plots in each vineyard. Dispensers were hung on every vine, however, row and vine spacing varied among vineyards (1.8 ×3.7, 2.1 × 3.7, 2.4 × 3.7, and 2.1 × 3.3 m) and, for this reason, dispensers were deployed at 935, 985, 990 and 1040 per ha (about 400 per acre), to create treatment rates of about 20 and 60 g (AI) / ha (8 and 24 g / acre). Each site had considerable Pl. ficus populations and 3 of 4 vineyards received an application of buprofezin on either 19 or 26 May or 5 July, and in 2 of 4 vineyards a post-harvest application of chlorpyrifos was applied on either 9 or 11 October.

to

There was a 20 row buffer between plots in each vineyard. Dispensers were hung on every vine, however, row and vine spacing varied among vineyards (1.8 ×3.7, 2.1 × 3.7, 2.4 × 3.7, and 2.1 × 3.3 m) and, for this reason, dispensers were deployed at 935, 985, 990 and 1040 per ha, to create treatment rates of about 20 and 60 g (AI) / ha. Each site had considerable Pl. ficus populations and 3 of 4 vineyards received an application of Applaud® (Nichino America, Wilmington, DE, USA) on either 19 or 26 May or 5 July, and in 2 of 4 vineyards a post-harvest application of Lorsban® (Dow AgroSciences LLC, Indianapolis, IN, USA) was applied on either 9 or 11 October

Note too that we have removed any discussion of dispensers per acre throughout the manuscript, as requested

7) change ml to mL – we have done this throughout the manuscript

8) Figure 4 caption was incomplete and we have changed the last part to

“(B) the categorical ranking of mealybug damage to fruit clusters at harvest-time across all treatments (different letters above each bar represent significant differences, pairwise comparisons using an experiment-wide error rate of P < 0.01.)” as requested.

Note that we have corrected this for Figures 5, 6, 9 and 10 as well.

9) In the Results it was suggested that we delete or move to M&M the comment “A major objective of the timed counts was to determine if MD treatments affected Pl. ficus developmental stages or locations on the vine.” We have moved this to the M&M.

10) We have rearranged the position of Fig 11 to bring the text and graphics into better perspective.

11 and 12) The Reviewer suggested a reduction or elimination of the first paragraph in the Discussion and so we have deleted all but the first sentence, and the Reviewer suggested that the conclusion should have no references and we have deleted those and made the Conclusions more concise.

Reviewer 2 Report

This is a 4-years field study on the effectiveness of MD for pest management of vine mealybug in California vineyards. The effects of MD on male captures, VM numbers and age structure on vines, was well as on damage to grape clusters were determined. The effect of formulation, dispenser load, release rate of pheromone dispenses were also assessed.  Although other studies have been published on MD of VM, this is the first extensive study presenting novel information on the effect of MD formulations and pheromone dose, as well as on the impact of MD on biological aspects of the populations of the VM.  Furthermore, the effectiveness of MD was evaluated in relatively high population densities of VM, in which MD is usually considered ineffective. Therefore, data presented can be considered as a reference of the effectiveness of MD in such unfavourable conditions.   The manuscript is well written, and presented, supported by updated literature. The research and analysis is scientifically sound. I would like to congratulate the authors for the work done. 

I recommend the ms could be accepted for publication after minor revision.

Author Response

(The authors gave the same response as above.)

Reviewer 3 Report

This manuscript investigated the "Development of a Mating Disruption Program for a Mealybug, Planococcus ficus, in Vineyards". The experimental set up of this study appears to be well-designed and the data collected carefully. However, some text problems need to be resolved. Revision could be checked in PDF attached.

Author Response

(The authors gave the same response as above.)
